# Relevance of Dynamic ^18^F-DOPA PET Radiomics for Differentiation of High-Grade Glioma Progression from Treatment-Related Changes

**DOI:** 10.3390/biomedicines9121924

**Published:** 2021-12-16

**Authors:** Shamimeh Ahrari, Timothée Zaragori, Laura Rozenblum, Julien Oster, Laëtitia Imbert, Aurélie Kas, Antoine Verger

**Affiliations:** 1Université de Lorraine, IADI, INSERM, UMR 1254, F-54000 Nancy, France; shamimeh.ahrari@univ-lorraine.fr (S.A.); timothee.zaragori@univ-lorraine.fr (T.Z.); julien.oster@inserm.fr (J.O.); l.imbert@chru-nancy.fr (L.I.); 2Sorbonne Université, AP-HP, Hôpitaux Universitaires Pitié-Salpêtrière Charles Foix, Service de Médecine Nucléaire and LIB, INSERM U1146, F-75013 Paris, France; laura.rozenblum@aphp.fr (L.R.); aurelie.kas@aphp.fr (A.K.); 3Department of Nuclear Medicine & Nancyclotep Imaging Platform, Université de Lorraine, CHRU-Nancy, F-54000 Nancy, France

**Keywords:** DOPA PET, glioma, recurrence, dynamic, radiomics

## Abstract

This study evaluates the relevance of ^18^F-DOPA PET static and dynamic radiomics for differentiation of high-grade glioma (HGG) progression from treatment-related changes (TRC) by comparing diagnostic performances to the current PET imaging standard of care. Eighty-five patients with histologically confirmed HGG and investigated by dynamic ^18^F-FDOPA PET in two institutions were retrospectively selected. ElasticNet logistic regression, Random Forest and XGBoost machine models were trained with different sets of features—radiomics extracted from static tumor-to-background-ratio (TBR) parametric images, radiomics extracted from time-to-peak (TTP) parametric images, as well as combination of both—in order to discriminate glioma progression from TRC at 6 months from the PET scan. Diagnostic performances of the models were compared to a logistic regression model with TBR_mean_ ± clinical features used as reference. Training was performed on data from the first center, while external validation was performed on data from the second center. Best radiomics models showed only slightly better performances than the reference model (respective AUCs of 0.834 vs. 0.792, *p* < 0.001). Our current results show similar findings at the multicentric level using different machine learning models and report a marginal additional value for TBR static and TTP dynamic radiomics over the classical analysis based on TBR values.

## 1. Introduction

Amino-acid PET radiotracers, such as 3,4-dihydroxy-6-[18F]-fluoro-L-phenylalanine (^18^F-FDOPA), are particularly useful for diagnosis of glioma recurrences [1,2,3], specifically high-grade gliomas (HGG) [4]. This is one of the underlying reasons why RANO (Response Assessment Neuro-oncology Group) has recommended assessment of gliomas using amino-acid PET radiotracers, in combination with MRI [5]. Indeed, one of the main limitations of conventional MRI is its inability to accurately differentiate glioma progression from treatment-related changes (TRC), given the relatively similar contrast enhancements observed in the two entities [5].

Amino-acid PET imaging in neuro-oncology is currently a fast-growing field, with diagnostic performances enhanced by dynamic [6] and/or radiomic [7] analyses. Radiomics, which involves extracting large amounts of image features, including morphological, statistical and textural features to characterize tumor heterogeneity, have not been widely studied in the context of glioma recurrence. To date, very few studies have investigated whether amino-acid PET-integrating radiomic analyses can differentiate glioma progression from treatment-related changes [8,9,10]. These studies did, however, show that radiomics could yield high diagnostic performances in this field, though none investigated ^18^F-FDOPA at a multi-centric level, directly comparing its performance to the current clinical standard of PET imaging (i.e., as opposed to classical tumor-to-background (TBR) parameters used in routine practice). The integration of dynamic PET imaging added considerable predictive value to conventional static parameters in terms of the initial diagnosis of glioma [6,11]. It is nevertheless noteworthy that this predictive value could not be extended to glioma recurrences, at least based on data from the currently available literature [3,12,13]. Indeed, our team recently showed, in a single-center ^18^F-DOPA PET study, that performances of dynamic parameters to differentiate glioma progression from treatment-related changes were lower than those of conventional static parameters in a population of mixed low-grade and high-grade gliomas (respective accuracies of 77% and 96% [3]). Recent studies have also reported better diagnostic performances for radiomic features obtained from dynamic parametric images as compared to more conventional dynamic parameters extracted from volumes of interest (VOIs) [12,13]. The same authors reported improved diagnostic performances of dynamic parameters extracted from the voxel level coupled with radiomic analysis compared to dynamic parameters extracted from a VOI, with the latter unable to predict the presence of TERT promoter mutation in gliomas at the initial diagnosis [12,14].

The current study therefore aims to evaluate the relevance of ^18^F-DOPA PET static and dynamic radiomic features, sourced from two independent nuclear medicine departments, for differentiation of HGG progression from that of TRC by assessing their diagnostic performances and comparing them to the current standard of PET imaging care.

## 2. Materials and Methods

### 2.1. Patients

To discriminate progression from TRC, we retrospectively identified patients with a histologically confirmed HGG investigated by dynamic ^18^F-FDOPA PET between November 2015 and June 2020, from two different institutions (CHRU of Nancy and Pitié-Salpêtrière hospital in Paris, France). All surgical tumor samples or stereotactic biopsies were classified according to the WHO 2016 classification [15]. To reduce the risk of ^18^F-FDOPA PET false positives, only patients with a minimum 3-month interval between the end of radiation therapy and the ^18^F-FDOPA PET acquisition were included. Final diagnoses were either determined from the histopathology or from the clinical-radiology follow-up during the 6-month follow-up period, based on the RANO working group criteria [5]. All patients included in the study gave their informed consent. The institutional ethics committee (Comité d’Ethique du CHRU de Nancy—FRANCE) approved the evaluation of retrospective patient data on 26 August 2020. The trial was registered at ClinicalTrials.gov (NCT04469244). The study complied with the principles of the Declaration of Helsinki.

### 2.2. PET Data Acquisition and Processing

All patients were asked to fast for at least 4 h prior to the PET scan, and some patients also received Carbidopa 1hr prior to their exam, depending on the procedural protocol in place at the respective centers. Following the injection of 2–3 MBq of ^18^F-FDOPA per kg of body weight, a 30-min dynamic PET acquisition was performed. Static PET images were reconstructed from the list mode data using the last 20 min of the acquisition. For dynamic PET images, 30 frames of 1 min each were reconstructed [11].

The PET images were obtained from four different imaging systems using locally optimized reconstruction parameters: (I) Biograph 6 True Point PET/CT (Siemens Healthineers^®^, Erlangen, Germany) with an OSEM 2D algorithm (2 iterations, 21 subsets, 256 × 256 × 148 voxels of 2.7 × 2.7 × 3.0 mm^3^) and 4-mm Gaussian post-reconstruction filter, (II) Vereos PET/CT (Philips Healthcare^®^, Eindhoven, The Netherlands) with an OSEM 3D algorithm (3 iterations, 15 subsets, 128 × 128 × 82 voxels of 2 × 2 × 2 mm^3^) and no post-filtering, (III) Biograph mCT Flow PET/CT (Siemens Healthineers^®^) with an OSEM 3D algorithm (8 iterations, 21 subsets, 400 × 400 × 109 voxels of 1.0 × 1.0 × 2.0 mm^3^) and no post-filtering, (IV) Signa 3T PET/MR (GE Healthcare^®^, Chicago, IL, USA) with an OSEM 3D algorithm (8 iterations, 28 subsets, 256 × 256 × 89 voxels of 1.2 × 1.2 × 2.8 mm^3^) and no post-filtering. Systems (I) and (II) are used in the CHRU of Nancy and systems (III) and (IV) in the Pitié-Salpêtrière hospital. Attenuation correction was performed using a 2-point Dixon MR sequence followed by a single atlas to capture bone information for the PET/MR [16], and the CT was used for PET/CT. No point-spread function correction was applied to any of the reconstructions.

### 2.3. Image Pre-Processing and Feature Extraction

To correct for the different voxel sizes of reconstructed images, all PET images were resampled into images with 2 × 2 × 2 mm^3^ voxels using the SimpleITK Python package [17] with a linear interpolation according to the Image Biomarker Standardization Initiative (IBSI) recommendations. Healthy brain and tumor VOI segmentations were performed by a nuclear physician (L.R.) using LifeX software (lifexsoft.org) [18], as previously described [11]. For healthy brain, a crescent-shaped VOI was positioned manually on three consecutive image slices on the semi-oval center of the unaffected hemisphere to include both white and gray matter [19]. Based on a threshold of 1.6 of mean standardized uptake value of healthy brain (SUV_mean_), a semi-automatic segmentation was used to determine tumor VOIs [19].

A correction for patient movements was performed on dynamic images to reduce any potential impact on voxel time-activity curves (TACs) due to long acquisition times. Dynamic images were registered on the CT for PET/CTs and on the MRI T1-enhanced gadolinium images for the PET/MR system. Moreover, working at a voxel level implies a greater influence of noise in TACs. Prior to TAC extraction, dynamic images were therefore denoised using the highly constrained backprojection local reconstruction (HYPR-LR) method, which has shown promising results for PET images [20]. As recommended in [20], PET images were denoised based on separate composites of uptake (frame 1:8), specific retention (frame 8:20), equilibrium (frame 20:30) and a 3D Gaussian with a FWHM of 9 mm.

Static images were normalized to the SUV_mean_ of healthy brain VOIs to neutralize the impact of carbidopa premedication on SUV measurements and to create static TBR parametric images. To avoid amplifying noise from TACs of dynamic images, voxel TAC_ratio_ was obtained by dividing the preliminary fitted voxel tumor TAC by the fitted mean brain TAC. These normalization methods were previously validated elsewhere [21]. Time-to-peak (TTP) values, which represent the time interval between tracer injection and the time point of the maximal TAC value, were extracted from each individual tumor at the voxel level to generate parametric TTP images.

### 2.4. Feature Extraction

From both static TBR and dynamic TTP parametric images, 94 radiomic features, including statistical, histogram-based, local-intensity and textural features were extracted using the tumor VOIs shared between the two types of images. Additionally, 11 common morphological features between the two image types were extracted. An absolute discretization of the images was performed with fixed bin sizes of 0.1 SUV and 1 min, respectively, for the static TBR and dynamic TTP parametric images, when required. To allow bins from different discretized images to be compared, the first bin was always designated as 0. For textural matrices, a 3D merging strategy was used [22], and only neighbors at a distance of 1 voxel were considered, with no distance weighting. To extract radiomic features according to the IBSI [22], the pyradiomics package was used (Available online: https://github.com/Radiomics/pyradiomics, accessed on 2 November 2021), as well as an in-house software for local-intensity features that were not available in pyradiomics [11]. Mathematical justifications of radiomic features have been given in [22]. To remove effects introduced by the use of different PET systems, features were harmonized with the modified ComBat method (Available online: https://github.com/Jfortin1/neuroCombat (accessed on 9 November 2021)) [23,24], with a digital Vereos PET device as a reference. Device effects were computed in a non-parametric manner using the empirical Bayes method to pool information across features. No biological covariates were considered. To investigate the effect of clinical data in combination with ^18^F-FDOPA PET parameters in the reference model, several clinical features, including age, sex, histopathological WHO grade, IDH mutation status, 1p/19q codeletion status, previous tumor resection and contrast enhancement on MRI were considered.

### 2.5. Model Building and Evaluation

Days of progression-free survival were dichotomized to a 6-month threshold and used as a reference label for the classification. To improve robustness and evaluate the general application of the learning algorithms, training and test sets were selected from different hospital centers. Patients from the CHRU of Nancy were used as training sets, and Pitié-Salpêtrière hospital patients were considered test sets. In the machine learning models presented below, all transformations and algorithms were fitted using only the training set and were subsequently applied to the test set.

All extracted radiomic features were initially normalized with z-score normalization. Dimensionality reduction was performed using hierarchical clustering based on an absolute spearman correlation coefficient (SCC) as distance matrix and a threshold of 0.9 [11,25]. These two previous steps were only performed on the numerical features before merging with the categorical features, where applicable. Due to class imbalance, the adaptive synthetic (ADASYN) sampling technique [26,27] was applied to oversample the minority class. Different machine learning algorithms were evaluated to identify robust comparisons: (I) ElasticNet logistic regression (LR), (II) random forest (RF) and (III) XGBoost (XGB) [28]. (I) and (II) were implemented in the scikit-learn Python package (Available online: https://scikit-learn.org/stable/index.html accessed on 15 November 2021), and (III) was implemented in the XGBoost Python package (Available online: https://xgboost.readthedocs.io/en/latest/index.html (accessed on 15 November 2021)). Each of the 3 models was trained with different sets of features: (I) radiomic features extracted from static TBR parametric images (94 static TBR radiomic features and 11 morphological features), (II) radiomic features extracted from TTP parametric images (94 TTP radiomic features and 11 morphological features), (III) a combination of (I) and (II) (94 static TBR radiomic features, 94 TTP radiomic features and 11 morphological features). As a previous study from our team demonstrated the high level of accuracy of VOI-based TBR_mean_ for prediction of glioma recurrences [3], three additional models were therefore fitted to serve as references: LR trained with (IV) previously mentioned clinical features, (V) TBR_mean_ and (VI) a combination of TBR_mean_ and clinical features.

The hyperparameters required for the different models were optimized appropriately. The main objective of hyperparameter tuning is to limit model overfitting and therefore also to better generalize on unseen data. The hyperparameters of the different learning algorithms were tuned only on the training set by applying an internal 5-fold cross validation (CV), which was repeated 20 times. The tuning process was driven by a Bayesian search based on optimization of Gaussian processes (Available online: https://github.com/scikit-optimize/scikit-optimize (accessed on 16 November 2021)), as it showed better results than a classical grid search and a random search [29]. The intervals and distributions for sampling sets of hyperparameters are provided in Table 1. The best hyperparameter set was the one yielding the minimal cross-entropy loss over the 300 iterations of the Bayesian search. Using the optimized hyperparameter set, 1000 models were trained on the training set using 1000 bootstrap iterations. For each bootstrap iteration, out-of-bag samples corresponding to the training samples that were not used to train the bootstrapped model were used to get a generalized performance on the training set that could also be considered a model validation. For each bootstrap, the trained models were then individually applied to the test set. Model performance on the test set was assessed based on different metrics to get a reliable mean generalized performance. The whole pipeline is summarized in Figure 1.

### 2.6. Statistical Analysis

Categorical variables are expressed as percentages, and continuous variables are expressed as means (range). Spearman correlation coefficients were used to compute correlations between TBR_mean_ and radiomic features from either TBR static or TTP dynamic parametric images. Diagnostic performances were determined from bootstrapped training samples, out-of-bag samples and testing samples using accuracy, area under the curve (AUC), precision, F1 score and balanced accuracy. On each set, the 95% confidence intervals (CI) of individual metrics were derived from the distribution of performances obtained with the individual 1000 bootstrapped, trained models. Unilateral comparisons of superiority were performed using Wilcoxon tests between the 1000 available AUCs, obtained from the predictions of the 1000 bootstrapped models on the test set for different models. Corrections for multiple comparisons [30] were applied. A *p*-value < 0.05 was considered significant. To evaluate the importance of features in each model, the static/dynamic dataset was assessed using Shapley additive explanations (SHAP) [31] on the test set. All analyses were conducted in Python (version 3.8.5; Available online: https://www.python.org/ (accessed on 2 November 2021)).

## 3. Results

### 3.1. Patient Characteristics

Ninety patients were initially retrospectively selected. Five patients were ultimately excluded to avoid mis-training of the models, while three patients had incomplete clinical information and dynamic images and data from two additional patients remained too noisy for voxel-based extraction of TTP, even after denoising. The final population therefore included 85 patients (average of 57 [21,80] years old, 46% women) with dynamic ^18^F-FDOPA PET acquisitions that could be considered for classification of a progression at 6 months from the PET scan. Seventy patients underwent a PET/CT exam, and 15 patients had a PET/MRI acquisition. The dataset was collected from two different centers. Data for 55 patients was obtained from the CHRU of Nancy, and the remaining 30 from the Pitié-Salpêtrière hospital (61 progressions at 6 months, 37 in data from Nancy and 24 in data from Paris). Fifty (59%) patients were premedicated with carbidopa. Tumor histopathology at initial diagnosis was either performed on tissue obtained during surgery (55 patients, 65%) or biopsy tissue (30 patients, 35%). According to the WHO 2016 classification of gliomas, eight (9%) patients were classified as having IDH-mutant anaplastic astrocytomas, 12 (14%) as having IDH-wildtype anaplastic astrocytomas, 10 (12%) as having IDH-mutant and 1p/19q anaplastic oligodendrogliomas, 6 (7%) as having IDH-mutant glioblastomas and 49 (58%) patients as having IDH-wildtype glioblastomas.

### 3.2. Correlation of Extracted Features with TBR_mean_

Figure 2 details the correlation coefficients of the TBR_mean_ and either: (a) morphological features, (b) radiomic features from static TBR parametric images or (c) radiomic features from dynamic TTP parametric images. Lower correlation coefficients were obtained for the reference TBR_mean_ feature and morphological features. In static TBR parametric images, some families of features (statistical, NGTDM, GLSZM and NGLDM) exhibited low correlation coefficients with TBR_mean_. A large number of features extracted from dynamic TTP parametric images were weakly correlated with TBR_mean_. All these findings suggest the potential added value of these parameters in the reference TBR model.

### 3.3. Classification of Progression at the 6-Month Follow-Up

The reference model for imaging features using LR trained with TBR_mean_ yielded a mean AUC value of 0.792 with a CI of 95% [0.792, 0.792] on the test set. Since the model was not complex and only involved one feature, the AUC value in the bootstrap analysis was the same (i.e., 0.792), which gave rise to a restricted 95% CI. The LR model trained with clinical features and with the combination of clinical features as well as TBR_mean_ gave AUCs of 0.670 [0.535, 0.757] and 0.789 [0.688, 0.847], respectively.

Details of the predictive performances for all the models, based on the different sets of features, are presented in Table 2. The AUC values for static, dynamic, and a combination of static/dynamic features based on LR were 0.715 [0.562, 0.799], 0.805 [0.597, 0.944] and 0.791 [0.618, 0.931], respectively, with the dynamic dataset yielding better results than the static model (*p* < 0.001). These results were confirmed by two other models: the RF and XGB. RF and XGB provided AUC values of 0.749 [0.576, 0.840] and 0.715 [0.535, 0.826] for static datasets, respectively, while the use of dynamic features led to AUC values of 0.832 [0.639, 0.965] and 0.755 [0.535, 0.910] (*p* < 0.001 for the comparisons between dynamic and static models). In addition, combining static and dynamic datasets led to respective AUC values for RF and XGB of 0.834 [0.674, 0.938] and 0.804 [0.646, 0.924] for each model (both superior to static models, *p* < 0.001). Using dynamic datasets for the three models combined or not combined with the static dataset only provided marginally added value relative to the reference TBR_mean_ model (AUC of 0.834 for the best machine learning model, i.e., the combination of static and dynamic datasets in the RF model, vs. AUC of 0.792 for the reference model, *p* < 0.001).

Introspection of the different models using SHAP values was provided for the static/dynamic datasets, as they include radiomic features of static TBR and dynamic TTP parametric images (Figure 3). For RF and XGB models, the TTP dynamic radiomic features gave values of the highest importance, with the 10th percentile from the statistics family and large-zone low-grey-level emphasis from grey-level size-zone matrix (GLSZM) being the two most influential contributive features. Although for the LR model, TBR static radiomic features were the most important, TTP dynamic radiomic features accounted for a large part of the model’s prediction capabilities. It appears that for all models, morphological and statistical features from both TBR static and TTP dynamic images, as well as texture matrices like GLSZM, neighborhood grey-tone difference (NGTDM) and grey-level run length (GLRLM) from TTP dynamic images, contributed more to the model.

All the extracted data are provided in a Appendix A.

## 4. Discussion

The current study highlights that radiomic features extracted from static TBR and dynamic TTP parametric images only provide slightly better performances in discrimination of HGG progression from TRC, compared to a simple model that only considers static TBR_mean_ parameters, as is currently performed in routine practice. This result was obtained by applying the robust radiomics method analysis in parallel with the current standards, using two independent training and testing patient datasets. Moreover, three different machine learning models were tested and led to the same results, thus strengthening the current findings.

Diagnostic performances for differentiation of HGG progression from TRC (AUC of 0.79 for the current reference model) are lower than those obtained in our previous work (AUC of 0.98 for the TBR_mean_ [3]), albeit within the range of values obtained in studies of large numbers of patients (AUC of 0.78 in a series of 110 patients with ^18^F-FDOPA PET [1] and of 0.75 in a series of 127 patients with ^18^F-FET PET for TBR_mean_ [10]). These lower performances obtained in our current work, when compared to our previous single-center study [3], may be related to a larger population size (85 patients vs. 51) and to the multi-centric nature of the present analysis. Interestingly, correlation analyses performed in the present study show that radiomic features from the morphological family, from several members of the family of static TBR parametric images and, to a more significant extent, radiomic features from TTP dynamic parametric images could provide significant additional value to the routinely used TBR_mean_ parameter, as confirmed by the low correlation coefficients between these radiomic features and the TBR_mean_ parameter (Figure 2). This justifies performing the present study to evaluate the added value of such radiomic features for differential diagnosis of HGG progression and TRC.

We previously reported that dynamic features extracted from a tumor VOI and radiomic features from static TBR parametric images were of added value for the prediction of molecular parameters at initial diagnosis of gliomas [11]. Our current results do not really replicate these findings for the prediction of recurrence in HGG. Results from our machine learning models only marginally outperform those of routine PET imaging based on the TBR_mean_ model. The latter has been defined as our reference since the addition of clinical features to this model did not show any significant diagnostic performance improvements (AUC of 0.79 for the combination of TBR_mean_ and clinical features, Table 2). In the context of differential diagnosis of glioma progression and TRC, dynamic parameters of amino-acid PET radiotracers, exclusively extracted from tumor VOIs, did not improve on the diagnostic performances of static parameters reported in the literature [3,32,33,34,35,36]. To date, few studies have investigated the value of amino-acid PET radiomic features in the context of glioma progression [8,9,10]. In a series of 34 glioblastoma patients, Lohmann et al. found that after increasing the number of ^18^F-FET PET scans to 102 by data augmentation, the reference TBR_mean_ model after ROC analysis gave an AUC of 0.73, similar to the AUC of 0.74 obtained with their machine learning model for diagnosis of glioma progression [10]. In a series of 160 gliomas, Wang et al. identified that a logistic regression model of static ^11^C-methonine PET radiomic features resulted in an AUC of 0.75 to differentiate glioma progression from TRC. These performances were increased to an AUC of 0.91 when ^11^C-methonine PET radiomic features were combined with those of ^18^F-FDG PET and contrast-enhanced MRI images [8]. However, the Wang et al. study did not include a comparison with a reference standard PET imaging model based only on SUV or TBR parameters. Carles et al. showed significant discrimination of progression-free survival in a series of 32 recurrent glioblastomas before repeat irradiation using Kaplan-Meier curves, but no C-index performances were reported, nor were comparisons to standard PET imaging TBR values included [9]. In contrast to the Wang et al. and Carles et al. studies, Lohmann et al. integrated dynamic parameters extracted from a VOI into their analyses [10]. To the best of our knowledge, our current study is therefore the first to include dynamic TTP parametric images to extract dynamic radiomic features to identify glioma recurrences. Interestingly, machine learning models integrating radiomic dynamic datasets systematically correlated with better performances than those only involving radiomic static datasets (Table 2). This is also confirmed by the greater importance attributed to radiomic features extracted from the dynamic TTP parametric image models trained with static/dynamic datasets (Figure 3). In addition, no other study has, to date, attempted to directly compare results obtained from radiomics machine learning models to those of conventional TBR static parameters also obtained from a machine learning process.

Radiomics extraction is a challenging and complex process that requires important steps in order to obtain accurate results. A meticulous methodological approach was performed to extract radiomic features according to the IBSI guidelines [22]. In addition, building machine learning models integrate crucial steps of feature normalization, dimension reduction [11,25] and corrections for oversampling [26,27]. Three different machine learning models were applied, i.e., LR, RF and XGB models, which all yielded similar results, thereby strengthening the fact that radiomic features only provide marginal additional value over a simple model involving only TBR_mean_ static parameters (Table 2). The SHAP values provided in Figure 3 confirm our results. Although these three machine learning models are based on different algorithms, very similar features or families of features are selected among the different models to build the optimized models (porphology family, statistics family from TBR static and TTP dynamic images, features extracted from textural matrices like GLSZM, NGTDM, GLRLM from TTP dynamic images for the three models). Importantly, and in contrast to the Lohman, Wang and Carles studies [8,9,10], our current study used radiomics on amino-acid PET imaging to identify glioma recurrences by training models on patient data from the center in Nancy, with external validation performed on different patient data sourced from Paris, which is an important criterion of robustness [37,38]. Moreover, it has been previously mentioned that this is a crucial aspect of reporting results for radiomic analyses, even if it leads to modest results, as is the case in the present study [39], i.e., only limited additional value of radiomic features over the conventional TBR parameter.

Our study suffers from several limitations. First, our population of HGGs included grade 3, as well as grade 4, gliomas, which may have opposed progression profiles, as would, for example, be expected for an anaplastic oligodendroglioma and an IDH-wildtype glioblastoma. Moreover, although we corrected for data harmonization with the modified Combat method, our study derived radiomic features from four different PET scanners using locally optimized acquisition and reconstruction parameters. Finally, our study did not identify any progression-free survival or overall survival benefits since radiomic features did not show significant added value over our conventional TBR parameter for our primary endpoint (progression at 6 months).

## 5. Conclusions

Radiomic features from static TBR and dynamic TTP parametric images only provide marginal additional value over a classical analysis based on TBR values for differentiation of HGG progression from TRC. These results are based on a robust machine learning analysis and may be of interest to nuclear physicians to limit the need to develop time-consuming routine radiomic PET imaging processes for this indication.

## Figures and Tables

**Figure 1 biomedicines-09-01924-f001:**
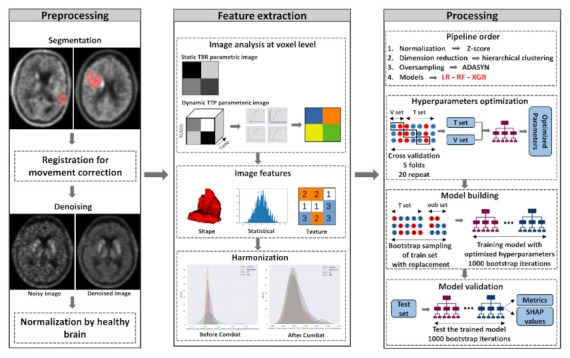
Pipeline summary. TBR: tumor-to-brain ratio, TTP: time to peak, ADASYN: adaptive synthetic, LR: ElasticNet logistic regression, RF: random forest, XGB: XGBoost, V set: validation set, T set: training set, oob set: out-of-bag set, SHAP: Shapley additive explanations.

**Figure 2 biomedicines-09-01924-f002:**
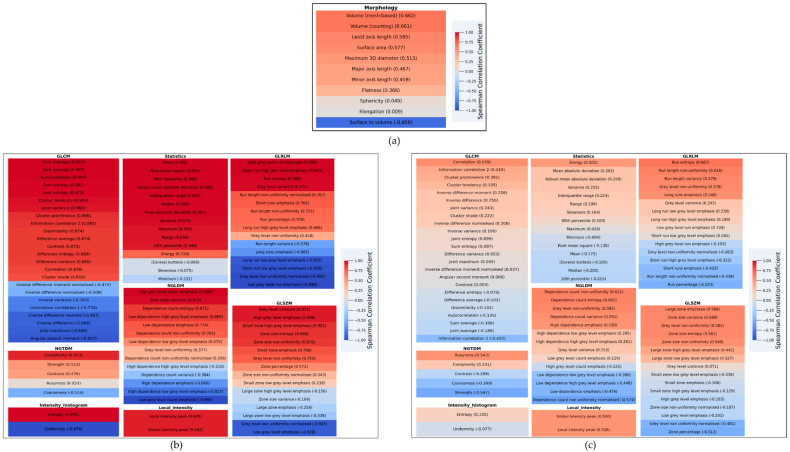
Heatmaps of correlation coefficients between TBR_mean_ and: (**a**) morphological features, (**b**) radiomic features from static TBR parametric images, (**c**) radiomic features from dynamic TTP parametric images. The features with light color show lower correlation coefficients with the TBR_mean_ feature, as this is the case for morphological features, a limited number of radiomic features from static TBR parametric images (statistical, NGTDM, GLSZM and NGLDM families) and a large number of features extracted from dynamic TTP parametric images. This information suggests the potential added value of these parameters in the reference TBR model.

**Figure 3 biomedicines-09-01924-f003:**
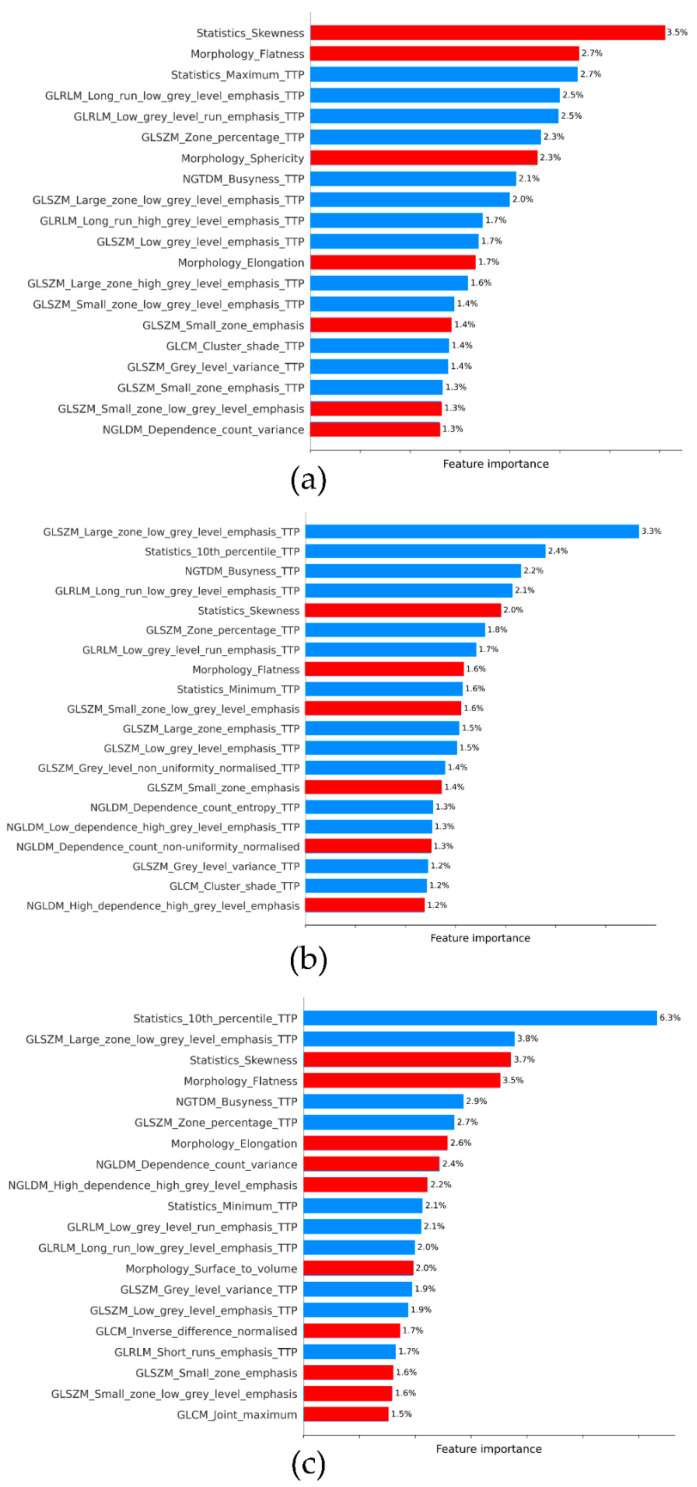
Representation of feature importance based on SHAP values for combination of radiomic features of static TBR and dynamic TTP parametric images: (**a**) ElasticNet logistic regression, (**b**) random forest and (**c**) XGBoost. The red and blue bars correspond to the radiomic features from static TBR and dynamic TTP parametric images, respectively.

**Table 1 biomedicines-09-01924-t001:** Intervals and distributions used for hyperparameter optimizations in each applied model.

Model	Parameters											
**LR**	**l1_ratio**						**C**					
	[0, 1] uniform						[0.001, 1000] log uniform					
**RF**	**n_estimators**			**max_features**			**max_depth**			**min_samples_leaf**		
	[50, 1000] uniform			[0.001, 1] uniform			[1, 20] uniform			[0.001, 0.5] uniform		
**XGB**	**n_estimators**	**max_depth**	**min child_weight**	**max_delta_step**	**learning_rate**	**gamma**	**subsample**	**colsample_bytree**	**colsample_bylevel**	**reg_alpha**	**reg_lambda**	**scale_pos_weight**
	[50, 1000] uniform	[1, 10] uniform	[1, 10] uniform	[0, 20] uniform	[0.001, 1] log uniform	[1 × 10^−9^, 0.5] log uniform	[0.01, 1] uniform	[0.01, 1] uniform	[0.01, 1] uniform	[1 × 10^−9^, 1] log uniform	[1 × 10^−9^, 1 × 10^3^] log uniform	[1 × 10^−6^, 500] log uniform

LR: ElasticNet logistic regression; RF: random forest; XGB: XGBoost; l1_ratio: ratio of L1 regularization; C: inverse of regularization strength; n_estimators: number of trees in the model; max_features: number of features considered when looking for the best split; max_depth: maximum depth of the tree; min_samples_leaf: minimum number of samples to be at a leaf; min_child_weight: minimum sum of sample weight needed in a child; max_delta_step: maximum difference step allowed in tree’s weight estimation; gamma: minimum loss reduction required to make a further partition on a leaf node of the tree; subsample: ratio of randomly selected samples to train each tree; colsample_by_tree: ratio of randomly selected features to train each tree, colsample_bylevel: ratio of randomly selected features for each depth; reg_alpha: L1 regularization strength; reg_lambda: L2 regularization strength; scale_pos_weight: balancing of positive and negative weights.

**Table 2 biomedicines-09-01924-t002:** Accuracies, AUC, precisions, F1 values and balanced accuracies of each tested model among the different datasets. Results are expressed as mean value with 95% confidence interval based on bootstrap samples. AUC: areas under the curve, oob: out-of-bag samples, TBR: tumor-to-brain ratio.

References															
Features/Metrics	Accuracy			AUC			Precision			F1			Balanced Accuracy		
	Train	Oob	Test	Train	Oob	Test	Train	Oob	Test	Train	Oob	Test	Train	Oob	Test
**Clinical**	**0.741** [0.600, 0.873]	**0.597** [0.375, 0.789]	**0.541** [0.400, 0.667]	**0.810** [0.692, 0.917]	**0.637** [0.507, 0.843]	**0.670** [0.535, 0.757]	**0.848** [0.742, 0.943]	**0.745** [0.562, 1.000]	**0.869** [0.778, 1.000]	**0.794** [0.667, 0.901]	**0.662** [0.316, 0.846]	**0.631** [0.471, 0.780]	**0.736** [0.589, 0.863]	**0.585** [0.414, 0.775]	**0.600** [0.479, 0.729]
**TBR_mean_**	**0.651** [0.527, 0.800]	**0.647** [0.471, 0.824]	**0.640** [0.567, 0.767]	**0.736** [0.575, 0.872]	**0.737** [0.538, 0.938]	**0.792** [0.792, 0.792]	**0.824** [0.697, 0.933]	**0.828** [0.667, 1.000]	**0.933** [0.923, 0.947]	**0.700** [0.567, 0.831]	**0.692** [0.471, 0.857]	**0.722** [0.649, 0.837]	**0.672** [0.533, 0.809]	**0.669** [0.492, 0.846]	**0.713** [0.667, 0.792]
**Clinical + TBR_mean_**	**0.809** [0.691, 0.910]	**0.873** [0.471, 0.842]	**0.631** [0.467, 0.800]	**0.884** [0.778, 0.963]	**0.720** [0.518, 0.923]	**0.789** [0.688, 0.847]	**0.902** [0.800, 0.972]	**0.792** [0.611, 1.000]	**0.935** [0.833, 1.000]	**0.849** [0.746, 0.935]	**0.739** [0.500, 0.889]	**0.711** [0.556, 0.864]	**0.812** [0.685, 0.918]	**0.653** [0.467, 0.850]	**0.708** [0.542, 0.812]
**ElasticNet Logistic Regression**															
**Features/Metrics**	**Accuracy**			**AUC**			**Precision**			**F1**			**Balanced Accuracy**		
	**train**	**oob**	**test**	**train**	**oob**	**test**	**train**	**oob**	**test**	**train**	**oob**	**test**	**train**	**oob**	**test**
**Static**	**0.800** [0.673, 0.909]	**0.673** [0.450, 0.850]	**0.697** [0.567, 0.833]	**0.854** [0.755, 0.938]	**0.694** [0.516, 0.900]	**0.715** [0.562, 0.799]	**0.896** [0.806, 0.970]	**0.797** [0.625, 1.000]	**0.904** [0.833, 0.950]	**0.842** [0.735, 0.930]	**0.734** [0.476, 0.897]	**0.783** [0.667, 0.894]	**0.802** [0.683, 0.904]	**0.661** [0.471, 0.850]	**0.699** [0.562, 0.812]
**Dynamic**	**0.874** [0.782, 0.945]	**0.668** [0.444, 0.842]	**0.745** [0.500, 0.867]	**0.950** [0.898, 0.992]	**0.701** [0.512, 0.905]	**0.805 ****^¥‡^* [0.597, 0.944]	**0.951** [0.879, 1.000]	**0.790** [0.600, 1.000]	**0.919** [0.833, 1.000]	**0.901** [0.818, 0.960]	**0.732** [0.435, 0.889]	**0.820** [0.579, 0.917]	**0.883** [0.782, 0.959]	**0.653** [0.446, 0.844]	**0.741** [0.562, 0.896]
**Static + Dynamic**	**0.886** [0.800, 0.964]	**0.687** [0.450, 0.870]	**0.754** [0.567, 0.867]	**0.946** [0.889, 0.994]	**0.720** [0.523, 0.917]	**0.791** *^¥^* [0.618, 0.931]	**0.949** [0.875, 1.00]	**0.802** [0.600, 1.000]	**0.911** [0.842, 1.000]	**0.912** [0.838, 0.972]	**0.748** [0.471, 0.903]	**0.830** [0.667, 0.920]	**0.890** [0.794, 0.972]	**0.672** [0.469, 0.857]	**0.732** [0.583, 0.875]
**Random Forest**															
**Features/Metrics**	**Accuracy**			**AUC**			**Precision**			**F1**			**Balanced Accuracy**		
	**train**	**oob**	**test**	**train**	**oob**	**test**	**train**	**oob**	**test**	**train**	**oob**	**test**	**train**	**oob**	**test**
**Static**	**0.824** [0.727, 0.909]	**0.651** [0.444, 0.826]	**0.732** [0.567, 0.867]	**0.909** [0.839, 0.965]	**0.696** [0.520, 0.893]	**0.749** [0.576, 0.840]	**0.905** [0.833, 0.971]	**0.784** [0.615, 1.000]	**0.912** [0.833, 0.952]	**0.862** [0.783, 0.933]	**0.715** [0.471, 0.875]	**0.811** [0.683, 0.913]	**0.823** [0.726, 0.917]	**0.640** [0.464, 0.822]	**0.723** [0.562, 0.833]
**Dynamic**	**0.953** [0.891, 1.000]	**0.643** [0.458, 0.810]	**0.741** [0.600, 0.867]	**0.993** [0.979, 1.000]	**0.687** [0.514, 0.867]	**0.832 ****^¥^* [0.639, 0.965]	**0.980** [0.943, 1.000]	**0.758** [0.588, 1.000]	**0.929** [0.826, 1.000]	**0.964** [0.921, 1.000]	**0.722** [0.500, 0.857]	**0.817** [0.700, 0.917]	**0.954** [0.903, 1.000]	**0.609** [0.433, 0.792]	**0.751** [0.562, 0.917]
**Static + Dynamic**	**1.000** [1.000, 1.000]	**0.688** [0.476, 0.842]	**0.791** [0.667, 0.900]	**1.000** [1.000, 1.000]	**0.717** [0.531, 0.898]	**0.834 ****^¥^* [0.674, 0.938]	**1.000** [1.000, 1.000]	**0.769** [0.600, 0.929]	**0.910** [0.833, 0.957]	**1.000** [1.000, 1.000]	**0.766** [0.526, 0.889]	**0.862** [0.780, 0.936]	**1.000** [1.000, 1.000]	**0.642** [0.458, 0.823]	**0.744** [0.583, 0.875]
**XGBoost**															
**Features/Metrics**	**Accuracy**			**AUC**			**Precision**			**F1**			**Balanced Accuracy**		
	**train**	**oob**	**test**	**train**	**oob**	**test**	**train**	**oob**	**test**	**train**	**oob**	**test**	**train**	**oob**	**test**
**Static**	**0.845** [0.745, 0.927]	**0.691** [0.384, 0.857]	**0.793** [0.300, 0.867]	**0.916** [0.850, 0.973]	**0.685** [0.518, 0.881]	**0.715** [0.535, 0.826]	**0.841** [0.725, 0.944]	**0.736** [0.571, 0.917]	**0.848** [0.786, 0.952]	**0.893** [0.831, 0.949]	**0.773** [0.286, 0.909]	**0.862** [0.222, 0.923]	**0.786** [0.611, 0.917]	**0.609** [0.433, 0.795]	**0.637** [0.479, 0.812]
**Dynamic**	**0.878** [0.800, 0.964]	**0.657** [0.389, 0.824]	**0.765** [0.633, 0.900]	**0.950** [0.893, 0.994]	**0.687** [0.513, 0.900]	**0.755** *^¥^* [0.535, 0.910]	**0.875** [0.787, 0.949]	**0.731** [0.571, 1.000]	**0.873** [0.792, 0.955]	**0.914** [0.857, 0.973]	**0.744** [0.333, 0.882]	**0.846** [0.744, 0.933]	**0.835** [0.722, 0.944]	**0.585** [0.400, 0.795]	**0.670** [0.479, 0.833]
**Static + Dynamic**	**0.859** [0.745, 0.964]	**0.687** [0.444, 0.842]	**0.823** [0.733, 0.900]	**0.980** [0.949, 1.000]	**0.709** [0.520, 0.900]	**0.804 ****^¥§^* [0.646, 0.924]	**0.833** [0.725, 0.949]	**0.710** [0.571, 0.867]	**0.843** [0.793, 0.917]	**0.906** [0.841, 0.974]	**0.786** [0.399, 0.897]	**0.894** [0.844, 0.941]	**0.786** [0.611, 0.944]	**0.566** [0.447, 0.750]	**0.624** [0.479, 0.792]

* *p*-value significant for the comparison with the reference TBR_mean_ model; *^¥^* *p*-value significant when compared to the static dataset using the same machine learning model; *^§^* *p*-value significant when compared to the dynamic dataset using the same machine learning model; *^‡^* *p*-value significant when compared to the static + dynamic dataset using the same machine learning model.

## Data Availability

Available in Appendix A.

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
