# Peer review of "Relevance of Dynamic 18F-DOPA PET Radiomics for Differentiation of High-Grade Glioma Progression from Treatment-Related Changes"

_biomedicines, 2021, doi:10.3390/biomedicines9121924_

Round 1

Reviewer 1 Report

The paper is very interesting and original. The potential impact of radiomics is emerging as huge factor and many papers are available in literature; instead, about DOPA PET in glioma no many evidences are available.

The article is well written, the methods seem to be correct and well-structured.

For this reason, the paper merits a high consideration.

However there are some points to clarify/discuss:

  • a crucial point not really clarify is the impact of scanner on features measuerements. We know that different scanners may affect features evaluation (please see doi: 10.3390/jcm10215064.). Besides, some of your scanes were acquired on a PET/MRI. Please study the role and impact of different scanners in radiomics measurements and compare if any differences are available.
  • in the abstract you described very well the methods and low the results. I changed the positions, more words about results and less about methods.
  • why didn't you evaluate PET classical variables? SUV? MTV? TLG? If possible, add the parameters in your analysis.
  • Figure 2 in very complex. Can youa dd something in the legend to explain better this figure?
  • ethical committee approval is lacking...

Reviewer 2 Report

In this interesting study the authors assessed the relevance of dynamic 18F-DOPA PET radiomics for differentiating high-grade glioma progression from treatment-related changes.

The methodology used by the authors is accurate and the results are well-described. Taking into account the findings reported in this article, radiomics features from static and dynamic 18F-DOPA PET images only provide marginal additional value over a classical analysis based on the tumor-to-background ratio values for differentiating high-grade glioma progression from treatment-related changes.

In my opinion radiomics is an hot topic in radiology and nuclear medicine and this article may be of significant interest for nuclear physicians and radiologists suggesting to not perform radiomics analysis for this indication.

Round 2

Reviewer 1 Report

The authors did not follow the suggestions done. They added only a prhase in a figure legend.

About ethical committe the protocol number of approvation is lacking.

MTV in like volumes of morphological features and TBR as SUV seems to be excessive

Round 3

Reviewer 1 Report

I suggest to accept the paper